# Ossification in Normal and Pathological Contexts: The Key Role of Static Osteogenesis vs. Dynamic Osteogenesis in the Etiopathology of Some Skeletal Alterations

**DOI:** 10.3390/biom15050733

**Published:** 2025-05-16

**Authors:** Carla Palumbo, Francesca Paganelli, Marzia Ferretti

**Affiliations:** 1Department of Biomedical, Metabolic and Neural Sciences, Section of Human Morphology, University of Modena and Reggio Emilia, 41124 Modena, Italy; marzia.ferretti@unimore.it; 2Department of Biomedical and Neuromotor Sciences, Alma Mater Studiorum, University of Bologna, 40136 Bologna, Italy; francesca.paganell16@unibo.it

**Keywords:** calcification, ossification, static osteogenesis, dynamic osteogenesis, skeletal pathologies

## Abstract

This Commentary is intended to start a discussion in the field of calcification/ossification related to osteogenesis. It highlights that two types of bone formation, static osteogenesis (SO) and dynamic osteogenesis (DO), are temporally followed by each other in bone histogenesis and bone lesion repair. Moreover, they also represent the common denominator in the pathological processes of both calcification and peculiar ossifications, such as heterotopic ossification and the formation of supernumerary skeletal segments. The final objective is to propose a different interpretation of certain bone alterations/pathologies, attributable to the two peculiar osteogenesis patterns (SO and DO), occurring in both physiological and pathological conditions. From these reflections, new approaches in the definition of diagnosis and therapies of certain alterations could be derived.

## 1. Introduction

In the various processes involving calcification, particularly ossification, under both physiological and pathological conditions (frequently associated with aging and/or different diseases), it is often possible to identify a common denominator. This is represented by the set of two processes, temporally successive to each other, which are named static osteogenesis (SO) and dynamic osteogenesis (DO), respectively [1].

Among the possible causes at the origin of pathological calcification/ossification events, just to give some examples of anomalies, are the following: (a) heterotopic ossification occurring in soft tissues in which bone does not normally form [2,3]; (b) supernumerary skeletal segments (together with body asymmetry, hypertrophy or hypotrophy) in both the axial and appendicular skeleton [4,5,6]; (c) segmentation defects during spine organogenesis [7]; (d) human facet joint osteoarthritis (FJOA), a degenerative spine disorder in aging [8]; (e) osteogenesis imperfecta (OI), an inherited disorder that prevents normal bone formation by making bones extremely brittle, caused by gene mutations [9,10].

As mentioned, these different types of anomalies share the same osteogenesis pathways.

## 2. Structural and Functional Differences Between SO and DO

Regarding osteogenesis, the first thing to state is that there is no unique process of bone formation. In fact, two different pathways of bone formation can occur on the basis of the following: (1) function to be performed, (2) time of occurrence, (3) types of osteoblasts involved (in relation to arrangement, polarization, motion), (4) conditioning factors to which different types of osteoblasts are sensitive (endothelial-derived cytokines vs. mechanical loading). It is important to underline that, both in physiological conditions and bone healing, SO and DO take place temporally one after the other during intramembranous ossification [1,11]. In particular, static osteogenesis is laid down by stationary osteoblasts (arranged in cords) and provides the preliminary rigid scaffold on which later dynamic osteogenesis, laid down by movable osteoblasts (arranged in laminae), produces mature bone tissue capable of meeting both the mechanical and metabolic skeletal needs (Figure 1). As a consequence of the two types of osteogenesis, the histology of the derived bone tissue is very different: woven-fibered bone containing haphazardly distributed globous osteocytes in SO and lamellar bone containing almond-shaped osteocytes ordered in planes in DO (Figure 2).

An additional diversifying aspect between SO and DO is the speed at which they take place: in SO the bone matrix is generally produced very rapidly, while in DO the events proceed over a longer period of time. Thus, the former rapidly allows for the production of a preliminary network of trabecular woven bone (surrounding wide primitive vascular spaces), which has a supporting function for subsequent lamellar bone apposition that needs more time to reach more organized bone texture [1,11].

The functional meaning of the two types of osteogenesis in terms of resistance to mechanical load is also relevant. SO produces bad-quality bone, whereas DO produces good-quality bone that is resistant to mechanical loading. These different features depend on both bone cellularity and the collagen arrangement of the bone matrix: high cellularity in woven texture by SO and fewer osteocytes located in planes in lamellar texture by DO [11] (Figure 3).

Another point worthy of discussion is the role of SO/DO in endochondral ossification, in order to understand whether the occurrence of SO and DO represents a universal osteogenic principle or is restricted to intramembranous contexts. In actual fact, in our specific study on endochondral ossification, SO never seems to take place [12]. Indeed, the osteoblasts in contact with the remnants of calcified cartilage are directly arranged in movable laminae and all appear to be functionally polarized in the same direction (i.e., toward the calcified cartilage). Moreover, the osteocytes inside the bone surrounding the calcified cartilage are never grouped inside confluent lacunae. This means that, in endochondral ossification, dynamic osteogenesis is not preceded by static osteogenesis. Hence, these observations on endochondral ossification confirm the hypothesis that dynamic osteogenesis needs a rigid mineralized surface to occur and that static osteogenesis only occurs in soft tissues where a rigid framework is lacking. This aspect is also in line with more recent observations by other authors [13,14].

## 3. Abnormal Processes of Ossification/Calcification

Considering the cases in which abnormal processes of ossification or calcification can occur, as far as heterotopic ossification (HO) is concerned, in a 2008 [2] description of a clinical case of HO in the scapulo-humeral region concomitant with keloid formation, we clearly demonstrated from the simple histological observations of the ectopic tissue (texture organization and shape/distribution of the cells) that the pattern of HO formation retraces the ontogenetic steps that normally occur along with intramembranous ossification: formation of woven-fibered bone by SO, which is the first to be formed, constituting the core of primary spongiosa, and lamellar bone later laid down by DO on primary core [2] (Figure 4). The same evidence was later reported by Ranganathan and colleagues in 2015 [3], where the authors indicated four classes of islands of HO within soft tissues of the hip, including the “early” histological heterotopic ossification corresponding to onset of bone tissue formation of our SO, with respect to more “mature” evolution of HO corresponding to successive formation by means of our DO. As far as the etiology is concerned, although the onset of the various forms of HO is still unclear, several authors suggest taking into consideration inflammatory triggers and the tissue environment, both closely related to the vascular context [15,16].

Abnormal ossifications may also occur in association with very peculiar pathologies like cutis marmorata telangiectatica congenita, whose pathophysiology is unknown, though it is proposed by some authors to be autosomal-dominant with low penetrance and multifactorial [17,18]. In a case report of cutis marmorata telangiectatica congenita [4], a supernumerary bone segment was observed in the skeleton of the foot; the abnormal morphological aspect of the supernumerary metatarsal bone can be explained by the up- or down-regulation of SO and DO, which affect the diaphyseal cortex, resulting in the bone being both thicker and with an osteoporotic-like aspect, due to enormous cavities which impart a spongy architecture to “compact” bone. In particular, the larger external diaphyseal size is based on the fact that the periosteal bone apposition, due to the succession of SO (first) and DO (later), was more intense than normal; in parallel, the endosteal bone resorption did not occur. As a consequence, the cortex of the supernumerary metatarsal bone was found to be incomparably thicker than normal, as result of peculiar modeling; this fact is in line with the limb hypertrophy. Moreover, the osteopenic appearance of the diaphyseal cortex is due to an unbalanced remodeling, viz, the fact that the first stage of bone remodeling (i.e., bone resorption) not only took place in an abnormal overwhelming manner but was also not followed by the successive stage of bone remodeling (i.e., bone deposition), normally occurring by DO. As a result, the cortex of the supernumerary metatarsal achieved a trabecular osteoporotic microarchitecture. Obviously, other causes could explain the abnormal development of supernumerary metatarsal bone, such as alterations of the angiogenesis of the metaphyseal bones as observed by Pasteels and colleagues [19].

Regarding segmentation defects during spine organogenesis [7], alterations of ossification can depend on multiple factors, some of which occur very early, such as disruption or injury to (i) the somitic mesoderm during gastrulation, (ii) the somites during segmentation or (iii) the sclerotomal precursors during the membranous phase that could unilaterally decrease the ability of the sclerotome to contribute to the formation of the vertebra. Although disordered ossification was originally proposed as a cause, the presence of these malformations in embryos from 7 to 11.5 weeks of gestation suggests that ossification is likely affected only at a later time [20,21,22]. In agreement with this suggestion, other authors observed that the failure of bone formation can be due to a deficiency of ossification due to a lack of vascularization [23]. In this regard, we have shown the onset of SO to be closely related to the location of blood vessels and to the presence of endothelial-derived cytokines, which are conditioning factors to which stationary osteoblasts are sensitive [1,11].

As far as human facet joint osteoarthritis (FJOA) is concerned, this pathology is relevant among the degenerative spine disorders and it is highly prevalent in aging populations, and considered a major cause of chronic lower back pain. In FJOA, the remodeling of the subchondral trabecular bone compartment is characterized by a peculiar increase in trabecular number [8]; this observation can be explained with the formation of new trabeculae by the recruitment of osteoprogenitor cells by endothelial-derived growth factors, which typically trigger static osteogenesis (in contrast to mechanical stresses which trigger dynamic osteogenesis). The same authors also commented that a histological analysis of osteocyte lacunae in healthy and osteoarthritic specimens could provide further support for the role of static osteogenesis in the pathogenesis of FJOA.

Last but not least, osteogenesis imperfecta (OI), a rare genetic syndrome involving skeletal fragility and increased exposure to bone fractures [24,25], is another pathology where static and dynamic osteogenesis explain some of the evidence, as described by Shapiro et al. in 2020 [9]. Our definitions of “stationary osteoblasts” in static osteogenesis and “movable osteoblasts” in dynamic osteogenesis correspond to the terms “MOBLs” and “SOBLs” previously used by Shapiro [10], in relation to the differences in both location and function of “mesenchymal osteoblasts” (MOBLs), which produce woven bone, with respect to “surface osteoblasts” (SOBLs), which produce lamellar bone. In fact, after the production of woven bone by MOBLs, SOBLs continue to secrete lamellar bone near the woven scaffold. Shapiro and colleagues [9] observed in OI that “the more severe the variant of OI is, the greater the persistence of woven bone and the more immature the structural pattern; the pattern shifts to a structurally stronger lamellar arrangement once a threshold accumulation for an adequate scaffold of woven bone is has been reached”. Similarly, in our review in 2021 [11], we correlated SO and DO with the deposition, in succession, of woven (first) and lamellar (later) bone by MOBL and SOBL, respectively, which occurs in both normal bone formation and repair of injured or pathological bone. It is interesting to note that in the less serious variant of OI, the organization of lamellar bone increases, as a result of increased dynamic osteogenesis.

## 4. Conclusions

To conclude, it is to be underlined that during histogenesis or the remodeling of hard tissues, physiological calcification is strictly related to the two processes of osteogenesis (SO vs. DO), also differently involved in various anomalies of ossification and bone pathologies. While various differences between the two types of osteogenesis are already well known (like conditioning factors, speed of occurrence, cellularity, texture and mechanical properties), the correlated process of calcification is yet to be investigated, and we are currently studying this intensively to understand whether the signaling pathways underlying calcification are similar or different in static vs. dynamic osteogenesis. Actually, calcification as well as bone mass is affected by mechanical loading; as previously reported, in bone formation and healing the first stages of bone deposition produce poor-quality bone due to inductive stimuli (likely of vascular origin), so that loading appears to be useless or sometimes even dangerous during static osteogenesis. On the contrary, mechanical loading, which is known to greatly enhance movable osteoblast activity during DO, becomes very important soon after the end of the process of bone formation by SO.

Among the previously described differences between the two types of osteogenesis, as far as the speed of occurrence is concerned, SO and DO differ in the fact that in SO the bone matrix is generally produced very rapidly, to allow the formation of a preliminary network of trabecular woven bone, surrounding the vascular spaces; as mentioned before, this means that the main purpose of SO is precisely to provide a preliminary rigid lattice to serve as a temporary support for the successive more ordered (viz, formed by lamellar bone) and slower deposition that characterizes DO. Thereafter, thickening of the SO-trabeculae by DO occurs and, as a result, narrowing of the primitive vascular spaces, giving rise to the primary osteons which will subsequently begin to undergo lifelong remodeling processes to adapt, moment by moment, the structure of the bone to the current metabolic and mechanical demands of the skeleton.

It could be relevant to explore the circumstances, if any, in which the calcification process follows different modalities in the two types of physiological osteogenesis and in the alterations of ossification, regardless of whether they concern heterotopic ossifications, supernumerary segments and/or anomalies secondary to altered morphogenesis processes during skeletal organogenesis.

The authors are aware of the complexity of the topic, but hope that the present “Commentary” can initiate and trigger a profound reflection on the role that calcification (often, in the context of skeletal tissues, trivialized as a routine event that follows the production of osseous matrix) may play in the etiopathogenesis of certain bone pathologies or alterations of morphogenetic processes that lead to congenital or acquired anomalies.

The final aim of our efforts is to start a discussion in the field of osteogenesis-related calcification/ossification that can trigger ideas and dissemination of results by all those researchers who share this important topic, in order to propose, based on a different reading key, new therapeutic strategies for the conditions of alteration of these processes.

## Figures and Tables

**Figure 1 biomolecules-15-00733-f001:**
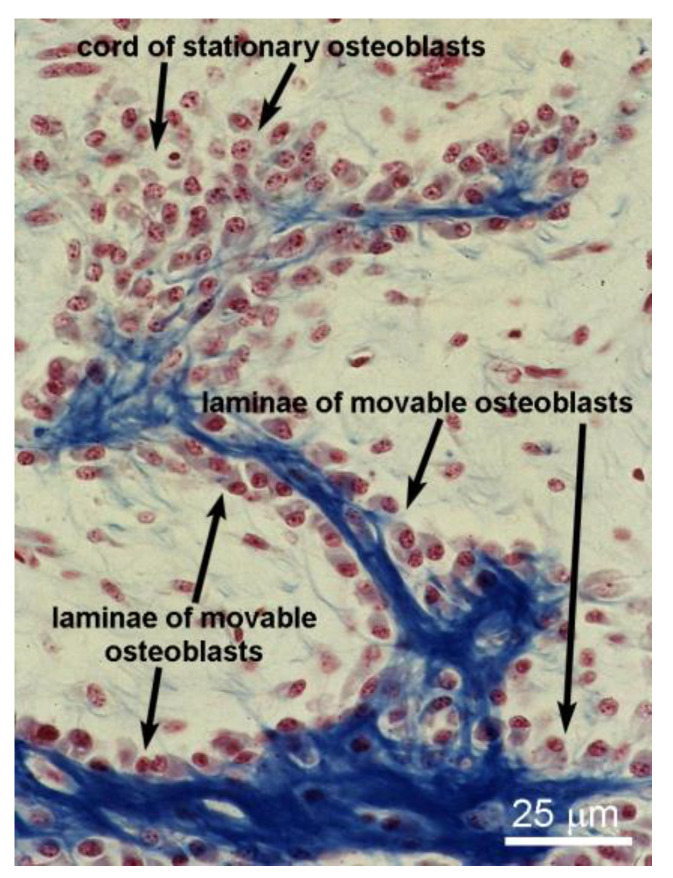
Histological section of an intramembranous ossification center showing a cord of stationary osteoblasts that provide the preliminary rigid scaffold (colored blue). Note the laminae of movable osteoblasts on the surfaces of pre-existing bone laid down by SO.

**Figure 2 biomolecules-15-00733-f002:**
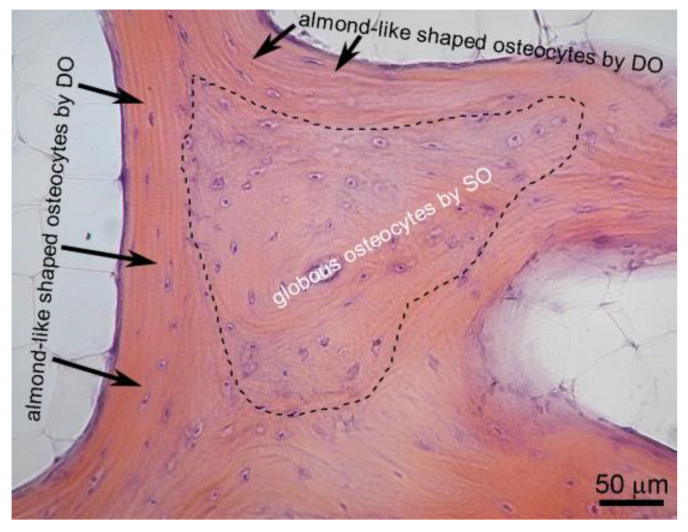
Histological section showing, as a result of the intramembranous ossification process, woven-fibered bone containing haphazardly distributed globous osteocytes (by SO) surrounded by the black dotted line, and lamellar bone containing almond-shaped osteocytes ordered in planes (by DO) indicated by black arrows.

**Figure 3 biomolecules-15-00733-f003:**
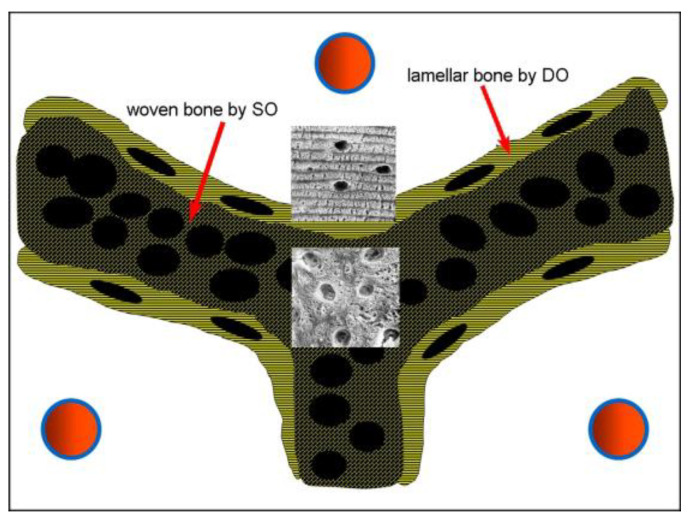
Schematic drawing of cellularity and texture in SO vs. DO. The core of preliminary trabeculae is formed by highly cellular woven bone by SO in which globous osteocyte lacunae are present. The lamellar bone by DO, surrounding the core, has fewer osteocyte lacunae of ellipsoidal shape located in planes. In the two inserts, SEM micrographs of lamellar bone (top) and woven bone (bottom) are shown. Red circles = vessels.

**Figure 4 biomolecules-15-00733-f004:**
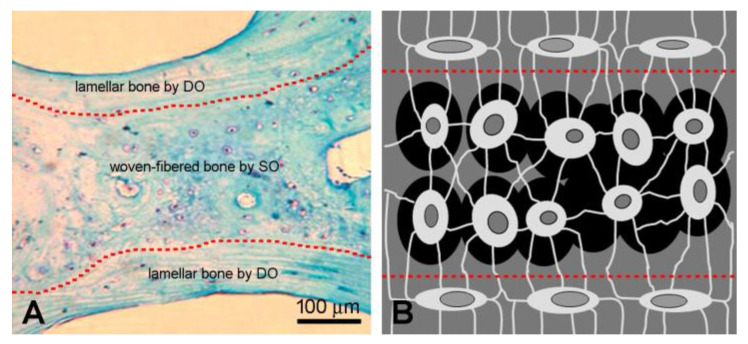
Histological section (**A**) of HO showing the two types of bone tissue coexisting in a trabecula: lamellar bone (by DO) covering both sides of the more deeply located woven-fibered bone (by SO) is delimitated by dotted red lines. In the two types of bone tissue, the microscopic arrangement is due to different disposition and morphology of osteocytes (globous or almond-shaped, sketched in (**B**)). (**B**): Globous osteocytes between the two red dotted lines; almond-shaped osteocytes above and below the lines.

## Data Availability

Data contained within the review are available in the papers cited in the References section.

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
