# Peer review of "Ossification in Normal and Pathological Contexts: The Key Role of Static Osteogenesis vs. Dynamic Osteogenesis in the Etiopathology of Some Skeletal Alterations"

_biomolecules, 2025, doi:10.3390/biom15050733_

Round 1

Reviewer 1 Report

Comments and Suggestions for Authors

The authors present an adequate commentary within the scope of this journal. It addresses an important topic and is well-focused. However, minor changes are needed:
1.The title should be a little more specific.
2.The figures are of good quality, but should be better described.
3.The figure legends should be better described.
4.The authors should include any clinical trials more specifically.
5.The authors should correct the grammatical errors.

Author Response

Comment 1: The title should be a little more specific.

Thank you for the suggestion. The authors agree and propose the following title integration: Ossification in normal and pathological contexts: the key role of static osteogenesis versus dynamic osteogenesis in the etiopathology of some skeletal alterations 

Comments 2 and 3: The figures are of good quality, but should be better described (2). The figure legends should be better described (3).

On the basis of reviewer’s suggestion, the authors have implemented the captions of figures 1, 2, 3 and 4, in order to clarify the description of the representative fields (see text).

Comment 4: The authors should include any clinical trials more specifically.

On the basis of this suggestion, the authors inserted some recent references on clinical aspects concerning the pathologies reported in the Commentary.

Comment 5: The authors should correct the grammatical errors.

Thank you very much. The authors red carrefully the manuscript and typos were corrected.

Reviewer 2 Report

Comments and Suggestions for Authors

The commentary provides a context in which to consider how bone is normally made as well as how different pathological processes may affect bone formation. This commentary builds from the review published by the authors in 2021 (reference 8).

Overall the commentary argues for a examining bone formation and pathologies from a more fundamental aspect of when and how bone formation is occurring rather than the phenotypic end product. As such, it is an interesting perspective.

Some minor points are lusted below. One point that should be clarified is the statement regarding the “eventual resolution of OI” (page 5 lines 159-161). OI encompasses a number of different mutations, each with varying effects on bone. As the mutations causing the OI cannot be cured, it is unclear what the authors meant be “resolution” and it is suggested that the authors expound upon or clarify this statement.

P2L41: change “no a single” to “no signal”

P2L44: typo iIV

P2L52: delete “also”

P4L89: typo organization

P5L153 and 156: suggested to use quote marks “…” instead <<…>>

Author Response

Comment 1: One point that should be clarified is the statement regarding the “eventual resolution of OI” (page 5 lines 159-161). OI encompasses a number of different mutations, each with varying effects on bone. As the mutations causing the OI cannot be cured, it is unclear what the authors meant be “resolution” and it is suggested that the authors expound upon or clarify this statement.

The authors sincerely thank the reviewer and apologise for the misunderstanding. It is true: the mutations that cause OI cannot be cured. The text has been changed to emphasise one aspect of the less severe forms of OI (see text).

Comment 2: P2L41: change “no a single” to “no signal”

The authors hypothesize a misunderstanding by the reviewer, probably due to an unclear term used by the authors. We have replaced the term ‘single’ with ‘unique’ to better explain that ‘bone formation does not result from a unique type of deposition process’.

Comment 3: P2L44: typo iIV

Thank you. The authors apologize: letters were replaced by numbers.

Comment 4: P2L52: delete “also”.

Correct. Thank you. Done.

Comment 5: P4L89: typo organization

Correct. Thank you. Done.

Comment 6: P5L153 and 156: suggested to use quote marks “…” instead <<…>>

OK, Done.

Reviewer 3 Report

Comments and Suggestions for Authors

This Commentary explores the pivotal roles of static osteogenesis (SO) and dynamic osteogenesis (DO) in physiological osteogenesis and various pathological calcification/ossification processes. It aims to trigger ideas about these two osteogenic patterns as foundational frameworks for understanding abnormal ossification or calcification. However, further in-depth exploration is required to enhance its scientific rigor and clinical signification.

  1. The content about the role of SO/DO in endochondral ossification is lack. This addition is necessary and critical because it explained whether the SO/DO represents a universal osteogenic principle or is restricted to intramembranous contexts.
  2. This Commentary explains the structural and functional differences between SO and DO, but the cellular and molecular biology mechanisms involved in these process are not specifically explained, which may be the molecular basis and intervention targets for pathological calcification.
  3. In addition to evidences from clinical sample analysis, relevant basic experimental research should be supplemented to support that the anomalies of ossification and bone pathologies is strictly related to the two processes of osteogenesis (SO versus DO).
  4. Based on the relationship between different osteogenic patterns (SO and DO) and pathological calcification, how to develop intervention strategies about relevant disease. Possible strategies or relevant content should be supplemented to highlight the importance and clinical significance of this Commentary.
  5. The References in the Commentary is relatively scarce for the past five years, lacking the content of the latest research advancements.
  6. Some sentences are excessively long, making them difficult to understand and failing to highlight their key points effectively, such as “line 9 to 13”, “line 23-27”.

Author Response

Comment 1: The content about the role of SO/DO in endochondral ossification is lack. This addition is necessary and critical because it explained whether the SO/DO represents a universal osteogenic principle or is restricted to intramembranous contexts.

Thank you very much for raising this thought. We have omitted this aspect, which however we have deepened and specifically investigated in a special study (Ferretti et al., Does static precede dynamic osteogenesis in endochondral ossification as occurs in intramembranous ossification? Anat. Rec., 288A: 1158-1162, 2006). In endochondral ossification, static osteogenesis never seems to take place; in fact, the osteoblasts in contact with the remnants of calcified cartilage are directly arranged in movable laminae and all appear to be functionally polarized in the same directions (i.e., toward the calcified cartilage). Moreover, the osteocytes inside the bone surrounding the calcified cartilage are never grouped inside confluent lacunae. This means that in endochondral ossification dynamic osteogenesis is not preceded by static osteogenesis. Hence, these observations on endochondral ossification support the hypothesis that dynamic osteogenesis needs a rigid mineralized surface to occur and that static osteogenesis only occurs in soft tissues where a rigid framework is lacking. This aspect has been introduced in the text of the Commentary, together with references of other authors who have confirmed later our observations.

Comment 2: This Commentary explains the structural and functional differences between SO and DO, but the cellular and molecular biology mechanisms involved in these process are not specifically explained, which may be the molecular basis and intervention targets for pathological calcification.

Thanks for the question. We have already started to investigate this aspect (sophisticated and therefore complicated to carry out) which, as mentioned in the first paragraph of the “Conclusions”, we plan to develop in the near future.

Comment 3: In addition to evidences from clinical sample analysis, relevant basic experimental research should be supplemented to support that the anomalies of ossification and bone pathologies is strictly related to the two processes of osteogenesis (SO versus DO).

The authors added new researches on the topics.

Comment 4: Based on the relationship between different osteogenic patterns (SO and DO) and pathological calcification, how to develop intervention strategies about relevant disease. Possible strategies or relevant content should be supplemented to highlight the importance and clinical significance of this Commentary.

Thanks for the question of suggesting possible strategies that is very important but … perhaps a little bit premature. The first thing to do according to the authors (which is why this Commentary was conceived) is to trigger a brain storming on the issue in order to share this key, and only afterwards, on the basis of the kind of feedback that will follow from various professionals, to share a forum where ideas and strategies can be sketched out.

Comment 5: The References in the Commentary is relatively scarce for the past five years, lacking the content of the latest research advancements.

Correct. Thank you. Done, the authors added recent researches.

Comment 6: Some sentences are excessively long, making them difficult to understand and failing to highlight their key points effectively, such as “line 9 to 13”, “line 23-27”.

Thank you. Done

Round 2

Reviewer 3 Report

Comments and Suggestions for Authors

My comments have been supplemented in the revised manuscript